# Features of SARS-CoV-2 Replication in Various Types of Reptilian and Fish Cell Cultures

**DOI:** 10.3390/v15122350

**Published:** 2023-11-29

**Authors:** Yulia Kononova, Lyubov Adamenko, Evgeniya Kazachkova, Mariya Solomatina, Svetlana Romanenko, Anastasia Proskuryakova, Yaroslav Utkin, Marina Gulyaeva, Anastasia Spirina, Elena Kazachinskaia, Natalia Palyanova, Oksana Mishchenko, Alexander Chepurnov, Alexander Shestopalov

**Affiliations:** 1Federal Research Center of Fundamental and Translational Medicine, The Federal State Budget Scientific Institution, Siberian Branch of the Russian Academy of Sciences, 2, Timakova St., Novosibirsk 630117, Russia; yuliakononova07@yandex.ru (Y.K.); aminisib@yandex.ru (L.A.); evgeniya.chubareva@gmail.com (E.K.); solomatina.mariyav@yandex.ru (M.S.); shestopalov2@mail.ru (A.S.); lena.kazachinskaia@mail.ru (E.K.); natalia.palyanova@gmail.com (N.P.); alexa.che.purnov@gmail.com (A.C.); stepanlomonosov@mail.ru (A.S.); 2Institute of Molecular and Cellular Biology, Russian Academy of Sciences, Siberian Branch, Novosibirsk 630090, Russia; rosa@mcb.nsc.ru (S.R.); andrena@mcb.nsc.ru (A.P.); utkinyaroslav99@gmail.com (Y.U.); 3The Department of Natural Science, Novosibirsk State University, 2, Pirogova St., Novosibirsk 630090, Russia; 448 Central Research Institute of the Ministry of Defense of the Russian Federation, Moscow 141306, Russia; 48cnii@mil.ru

**Keywords:** SARS-CoV-2, aquatic ecosystems, fish cells, reptile cells, TH-1 cell culture

## Abstract

Background: SARS-CoV-2 can enter the environment from the feces of COVID-19 patients and virus carriers through untreated sewage. The virus has shown the ability to adapt to a wide range of hosts, so the question of the possible involvement of aquafauna and animals of coastal ecosystems in maintaining its circulation remains open. Methods: the aim of this work was to study the tropism of SARS-CoV-2 for cells of freshwater fish and reptiles, including those associated with aquatic and coastal ecosystems, and the effect of ambient temperature on this process. In a continuous cell culture FHM (fathead minnow) and diploid fibroblasts CGIB (silver carp), SARS-CoV-2 replication was not maintained at either 25 °C or 29 °C. At 29 °C, the continuous cell culture TH-1 (eastern box turtle) showed high susceptibility to SARS-CoV-2, comparable to Vero E6 (development of virus-induced cytopathic effect (CPE) and an infectious titer of 7.5 ± 0.17 log_10_ TCID_50_/mL on day 3 after infection), and primary fibroblasts CNI (Nile crocodile embryo) showed moderate susceptibility (no CPE, infectious titer 4.52 ± 0.14 log_10_ TCID_50_/mL on day 5 after infection). At 25 °C, SARS-CoV-2 infection did not develop in TH-1 and CNI. Conclusions: our results show the ability of SARS-CoV-2 to effectively replicate without adaptation in the cells of certain reptile species when the ambient temperature rises.

## 1. Introduction

Coronavirus SARS-CoV-2 (family Coronaviridae, genus *Betacoronavirus*, subgenus Sarbecovirus) first appeared in December 2019 in China (Wuhan, Hubei Province) as the causative agent of unknown pneumonia. Later, it was the cause of the pandemic of a new coronavirus infection, COVID-19. The first human cases were visitors to a wholesale seafood market, which suggested that the source of the virus was exotic animals used in the local popular cuisine [1]. In early studies, the RaTG13 isolate from the intermediate horseshoe bat (*Rhinolophus affinis*) betacoronavirus, whose genome has 96.2% homology with the genome of the COVID-19 pathogen, was considered the most likely precursor of SARS-CoV-2 [2]. Based on the biological properties and phylogenetic analysis of betacoronavirus isolates from Sunda pangolins (*Manis javanica*), these animals were assumed to be the most likely intermediate hosts of SARS-CoV-2 [3]. New data on the spread of SARS-CoV-2 at the beginning of the pandemic and the pre-pandemic period [4,5], as well as an analysis of the genetic evolution of the COVID-19 pathogen and related betacoronaviruses in bats and pangolins [6], give grounds to consider different hypotheses about its origin.

During the evolution of the epidemic process, the new coronavirus SARS-CoV-2 has demonstrated the ability to overcome the interspecies barrier and be transmitted from humans to some mammals—domestic animals (cats, dogs, ferrets, minks, hamsters) and wild animals in zoos and in nature [7,8]. According to the World Organization for Animal Health (WOAH), as of the end of June 2023, cases of disease of varying severity caused by SARS-CoV-2 during natural infection or asymptomatic virus carriers have been documented in 29 mammalian species in countries of North and South America, Asia, Africa, and Europe [9]. In some episodes (mink on farms, domestic hamsters), cases of reverse zoonotic transmission of SARS-CoV-2 from infected animals to humans have been reported [10,11]. The transition of SARS-CoV-2 from the human population to a new host can lead to the formation of independent foci of virus circulation with unpredictable epidemic consequences.

Even at the beginning of the pandemic, it was found that SARS-CoV-2 RNA was present not only in smears from the upper respiratory tract of COVID-19 patients, but also in feces and urine [12,13]. In this regard, at the initial stage of the pandemic, assumptions were made about the possibility of fecal-oral transmission of SARS-CoV-2 [14,15], as well as of the virus entering the environment both from contaminated raw sewage [16] and directly from excrement [17]. Despite the high frequency of detection of viral RNA in the feces of those affected by COVID-19, cases of successful isolation of the infectious virus from feces and urine are infrequent [18,19,20,21]. Also, during the entire period of the COVID-19 pandemic, no cases of isolation of infectious SARS-CoV-2 from waste and river water samples containing viral RNA have been documented [22]. An extremely small proportion of infectious coronavirus isolates from feces, and their complete absence when trying to isolate the virus from waste and river waters, can be associated both with a low concentration of SARS-CoV-2 in such samples [23,24] and the loss of its infectious properties [25]. It is worth noting that when trying to isolate a live virus from feces, urine, and contaminated water, the laboratory procedures that have been most used are based on variants of Vero cell culture (African green monkey (*Chlorocebus sabeus*) kidney) [25,26], CaCo2 cell culture (human colorectal adenocarcinoma) [27,28], and Syrian hamster (*Mesocricetus auratus*) [26]. These may be less sensitive to testing such material than they are to upper respiratory swabs from COVID-19 patients with a high viral load. Jeong et al., using the example of intranasal infection of ferrets with virus-containing samples of feces and urine of patients with COVID-19, revealed the presence in the ferrets tested of infectious SARS-CoV-2, while infection of Vero cell culture with the same samples did not lead to the development of infection [21]. This indicates a greater sensitivity of ferrets as bioassays for such material and the possible occurrence of false negative results when analyzing the viability of SARS-CoV-2 in feces, urine, and water when using other laboratory cell culture systems. Thus, the presence of infectious SARS-CoV-2 in the feces of patients with COVID-19 and virus carriers, as well as the threat of the virus entering the aquatic environment, are likely underestimated.

The interaction of SARS-CoV-2 with animals in aquatic ecosystems is currently not well understood. As of May 2023, there are three reports of findings of SARS-CoV-2 RNA in the digestive systems of bivalve mollusks: Manila clam (*Ruditapes philippinarum*) and European carpet shell clam (*Ruditapes decussatus*) in Spain [29], European zebra mussel (*Dreissena polymorpha*) in France [30], and Black Sea mussels (*Mytilus galloprovincialis*) in Italy [31]. Mollusks are associated with aquatic vertebrates (fish) and coastal ecosystems (some amphibians, reptiles, mammals, and birds) through food chains. In such ecological situations, when infected mollusks are eaten, the probability of overcoming the interspecies barrier by the coronavirus increases after its proteolytic processing in the digestive system of animals that feed on them [32]. In addition to the bivalve mollusk species listed above, other poikilothermic animals of aquafauna and coastal ecosystems have not been tested for the presence of SARS-CoV-2 markers.

Cases of natural infection with SARS-CoV-2 have also been described in mammals with aquatic and semi-aquatic lifestyles: wild American mink (*Neovison vison*) tested in Spain [33], Asian small-clawed otter (*Lutra cinerea*) tested in the USA [34], Eurasian otter (*Lutra lutra*) tested in Spain [35], Antillean manatee (*Trichechus manatus manatus*) tested in Brazil [36], and hippopotamus (*Hippopotamus amphibius*) tested in Belgium and Vietnam [37,38]. If the animals among these that are kept in captivity (Asian small-clawed otter, Antillean manatee, hippopotamus) are infected, one can assume a route of infection that is not associated with water (e.g., from virus carriers among zoo personnel), while in the case of wild American mink and Eurasian otters, infection with SARS-CoV-2 would have occurred in their natural habitat. In the wild, American mink and river otters inhabit the coastal zones of rivers, lakes, canals, and reservoirs, obtaining food (fish, frogs, crustaceans, and mollusks) mainly in the water [39,40]. These animals belong to the marten family (Mustelidae) as ferrets, which are highly susceptible to SARS-CoV-2 infection. The lifestyle of these animals and the recorded high species susceptibility to SARS-CoV-2 do not exclude the possibility of their infection with coronavirus in an aquatic environment.

Currently, there are individual studies on the susceptibility to SARS-CoV-2 of poikilothermic vertebrates in vivo and in cell cultures. Resistance to SARS-CoV-2 was shown in XTC-2 cell culture obtained from the clawed frog (*Xenopus laevis*) [41], and three freshwater fish cell cultures—EPC (fathead minnow *Pimephales promelas*, skin), CIK (grass carp *Ctenopharyngodon idella*, kidney) and BF-2 (bluegill sunfish *Lepomis macrochirus*, caudal peduncle) [42]. SARS-CoV-2 infection of zebrafish larvae (*Danio rerio*) [43] and adults of three spot gourami (*Trichopodus trichopterus*) [44] did not lead to infection. Happi AN et al. detected SARS-CoV-2 RNA in oral and cloacal swab samples of common agama (*Agama agama*) in two states of Nigeria in 2021–2022 [45]. Data on the study of the susceptibility of reptiles to SARS-CoV-2 in vivo and their cells in vitro are not available in accessible sources.

The aim of this work was to study the species tropism of SARS-CoV-2 and the possibility of replication in various types (continuous, primary, epithelial-like, fibroblasts) of cell cultures of certain poikilothermic vertebrates: freshwater fish (fathead minnow, silver carp) and reptiles (eastern box turtle, Nile crocodile), as well as the effect of incubation temperature on this process.

## 2. Materials and Methods

### 2.1. Virus 

In this work, a 7th passage strain of the SARS-CoV-2/human/RUS/Nsk-FRCFTM-1/2020 nCoV strain (wild-type, original variant) was used from the Specialized Collection of Reference Cultures of Virus Strains of II-IV Pathogenicity Groups (Arboviruses and Others) of the 48 Central Research Institute of the Ministry of Defense of the Russian Federation. Virus production was carried out on Vero cell culture (African green monkey kidney) according to the method earlier described [46]. The infectious virus titer was determined by the final dilution method by infection with 10-fold dilutions of the Vero E6 cell culture (African green monkey kidney) in a 96-well plate according to [47], with calculation of the virus titer by the Reed-Muench method. All experiments were carried out in a laboratory with biosafety level 3 (BSL-3).

### 2.2. Cell Cultures

Continuous, primary, and diploid cell cultures of fish and reptiles were used in our experiments. The type of culture, species, organ (tissue) origin, cultivation mode, and collection are presented in Table 1.

To study the tropism of SARS-CoV-2 for the cells of some fish and reptile species, we used continuous cell cultures FHM and TH-1, primary fibroblasts CNI (4th passage), and diploid fibroblasts CGIB (22nd passage). The origin of CNI fibroblast culture was previously described by Romanenko et al. [48], and the CGIB diploid fibroblasts culture was obtained at the IMCB, SB RAS. Vero E6 cell culture susceptible to SARS-CoV-2 was used as a reference culture. Cultivation was carried out on nutrient media MEM, DMEM (Biolot, Moscow, Russia), and Alpha MEM (Gibco, New York, NY, USA), with Capricorn Scientific fetal bovine serum (South America) and gentamicin (PanEco, Moscow, Russia) in culture flasks with a surface area of 25 cm^2^ (TPP, Trasadingen, Switzerland).

### 2.3. Infection

The monolayer of FHM, TH-1, CNI, CGIB, and Vero E6 (90–100% confluence) was washed with a sterile Hank’s solution (Biolot, Russia). After washing, 100 μL of the SARS-CoV-2/human/RUS/Nsk-FRCFTM-1/2020 strain containing 3.7 log_10_ TCID_50_ (Dose 1) and 1 mL of the corresponding culture medium without FBS were applied to the monolayer and left for adsorption for 1 h at 25 °C, 29 °C, and 37 °C (Vero E6 only). Fish cell cultures were additionally infected with a higher dose of the virus, 5.7 log_10_ TCID_50_/100 µL (Dose 2). After adsorption, the virus-containing liquid was removed, the monolayer was washed with Hank’s solution, and a supporting medium corresponding to each cell culture was added with a suitable FBS content of 2% for TH-1, FHM, and Vero E6, and 4% for CNI and CGIB. After addition of the maintenance medium, culture flasks were incubated at 25 °C, 29 °C, and 37 °C (Vero E6 only). The selected temperature regime is acceptable for the study of viral infections in continuous cell cultures [49,50,51,52]. For CNI and CGIB fibroblast cultures, cell viability was assessed at selected temperatures prior to infection.

### 2.4. Study of SARS-CoV-2 Infection in Cell Cultures

Virus replication was assessed by the appearance of a cytopathic effect (CPE) and the presence of viral RNA and infectious virus in the culture medium in triplicate, in two independent experiments. The state of the cells was observed for 7 days from the moment of infection using an inverted microscope Mikromed I (Mikromed, Russia) at 10× magnification. For photodocumentation of the state of the monolayer, a ToupCam 5.1 MP video eyepiece (ToupCam, China) with accompanying ToupView 3.7 software was used. Sampling of the culture medium of infected cells with monolayer scraping (in the absence of CPE) for real-time polymerase chain reaction (RT-PCR) and determination of the infectious titer was performed on day 0 (immediately after the introduction of the supporting medium) and days 1, 2, 3, 5, and 7.

Isolation of RNA from the culture medium of infected cell cultures was performed using the RealBest Extraction 100 kit (JSC Vector-Best, Russia), according to the manufacturer’s instructions. The presence of SARS-CoV-2 RNA was determined using commercial assay “SARS-CoV-2 virus RNA detection system (N gene) kit” (Biolabmix, Russia), according to the manufacturer’s instructions. To determine the infectious titer of SARS-CoV-2, the final dilution method was used by infecting a Vero E6 cell culture in a 96-well plate with aliquots of the culture medium of infected cell cultures and their serial 10-fold dilutions [47], with the calculation of the virus titer according to Reed-Mench.

## 3. Results

### 3.1. SARS-CoV-2 Infection in Reptile Cell Cultures

State of monolayer. At 25 °C, infected TH-1 and CNI cells showed no change in monolayer state compared to mock. At a temperature of 29 °C in the cell culture of the Carolina box turtle TH-1, starting from the third day after infection, the development of CPE was observed with the appearance of foci of destruction of the monolayer, with progression up to complete destruction by the fifth day after infection (Figure 1, TH-1). In the culture of fibroblasts of Nile crocodile embryo CNI, the integrity of the monolayer was not disturbed during the observation period. In culture flasks with infected cells, by day 7, the accumulation of single detached cells in the culture medium was noted in comparison with mock (Figure 1, CNI).

SARS-CoV-2 RNA and infectious titer over time. At a temperature of 25 °C in infected TH-1, CNI cells, as well as in the Vero E6 comparison culture, no accumulation of SARS-CoV-2 RNA and infectious virus was observed during the entire experiment (Figure 2A,B). At 29 °C, TH-1, CNI, and Vero E6 cells showed a decrease in Ct (Cycle Threshold) values (accumulation of subgenomic RNA of the coronavirus N protein) (Figure 2C). In TH-1 and CNI cells, an increase in the infectious titer was observed, followed by a decrease from days 3 and 5, respectively. In the Vero E6 comparison culture, the infectious titer of SARS-CoV-2 increased from day 0 to day 7 without a decrease (Figure 2D).

### 3.2. SARS-CoV-2 Infection in Fish Cell Cultures

State of Monolayer. No changes were observed in the state of the monolayer in FHM and CGIB cells during the experiment, regardless of the dose of infection and temperature in comparison with mock.

SARS-CoV-2 RNA and infectious titer over time. As in the reptile cells, no accumulation of SARS-CoV-2 RNA was observed in infected FHM and CGIB fish cells at 25 °C, regardless of virus dose (Figure 3A). Infectious virus was not detected in the culture medium of fish cells infected with a dose of 3.7 log_10_ TCID_50_/100 µL (Dose 1) throughout the experiment. In FHM cell culture infected with 5.7 log_10_ TCID_50_/100 µL (Dose 2), infectious SARS-CoV-2 was present in the culture medium for up to 3 days, with a gradual decrease in titer. In CGIB diploid fibroblasts, infectious virus persisted in small amounts (0.89 ± 0.19 log_10_ TCID_50_/mL) immediately after adsorption (Figure 3B). At an incubation temperature of 29 °C, no accumulation of viral RNA was observed in infected FHM and CGIB cells (Figure 3C). Infectious SARS-CoV-2 was absent in the culture medium of cells infected with a dose of 3.7 log_10_ TCID_50_/100 µL. At a higher infecting dose immediately after adsorption in FHM cells, the titer of the infectious virus was significantly higher than at 25 °C (3.27 ± 0.25 log_10_ TCID_50_/mL versus 2.5 ± 0.17 log_10_ TCID_50_/mL); however, by the second day after infection, live SARS-CoV-2 was not detected. Susceptibility to SARS-CoV-2 of CGIB fibroblasts at 29 °C, infected with a dose of 5.7 log_10_ TCID_50_/100 µL, did not differ from that at 25 °C (Figure 3D).

## 4. Discussion

Currently, the Coronaviridae family includes three subfamilies: Orthocoronavirinae, Letovirinae, and Pitovirinae [53]. The largest subfamily, Orthocoronavirinae, includes four genera: alphacoronaviruses (*Alphacoronavirus*), betacoronaviruses (*Betacoronavirus*), deltacoronaviruses (*Gammacoronavirus*), and gammacoronaviruses (*Deltacoronavirus*). The first two genera are represented by mammalian coronaviruses (including SARS-CoV-2), delta- and gammacoronaviruses are found mainly in birds, and deltacoronaviruses are also found in some mammals. The subfamily Letovirinae contains only one genus, *Alphaletovirus*, and a subgenus, Milecovirus, with a single species, Microhyla letovirus 1, identified in the tree frog (ornate pigmy frog *Microhyla fissipes*) in China [54]. The subfamily Pitovirinae also contains a single genus, *Alphapironavirus*, and a subgenus, Samovirus, with a single Alphapironavirus bona species, which was found in sick Chinook salmon (*Oncorhynchus tshawytscha*) in Canada [55]. Miller et al., when analyzing the transcriptomes of marine and freshwater bony fishes and the Australian lamprey, identified nucleotide sequences that form a separate phylogenetic group closest to the subfamily Letovirinae [56]. In this regard, results obtained by several groups of researchers suggest a significant diversity of susceptible hosts for viruses of the Coronaviridae family.

Under natural conditions, vertebrate aquatic animals comprised of fish (Pisces), lampreys (Petromyzontida), hagfish (Myxini), and animals associated with coastal ecosystems (certain species of amphibians (Amphibia) and reptiles (Reptilia)) can come into contact with coronaviruses of all four Orthocoronavirinae genera (Figure 4A). In lampreys, hagfishes, fishes, and reptiles living in the waters of the seas and oceans, contact can occur with gammacoronaviruses of cetaceans (subgenus Cegacovirus), alpha- and gammacoronaviruses of seals [57], and in the coastal zone, contact can occur with gamma- and deltacoronaviruses of seabirds. Contact can also occur with human and animal alpha- and betacoronaviruses from wastewater (Figure 4B). Representatives of all genera of coronaviruses can also enter freshwater ecosystems with feces from natural hosts—bats and birds, from susceptible hosts among wild and domestic animals, and with wastewater from agricultural farms and settlements (Figure 4B). Coronaviruses thus introduced into freshwater ecosystems may interact with the animals that inhabit them, including fish and lampreys, as well as with amphibians and reptiles associated with aquatic systems through foraging or reproduction. The probability of introducing coronaviruses into aquatic systems increases markedly in the event of flashfloods, floods, and tsunamis, as well as natural and man-made disasters that lead to the destruction of water treatment facilities.

The SARS-CoV-2 also enters aquatic ecosystems, as evidenced by the detection of viral RNA in waste- and river waters in various regions of the world, both during the pandemic [16,23,24] and in the pre-pandemic period [5]. Findings of SARS-CoV-2 RNA in the digestive system of several species of bivalve mollusks [29,30,31], as well as in samples of the internal organs of wild American mink and river otters taken near rivers and reservoirs [33,35], indicate its interaction with aquatic animals and coastal ecosystems. Poikilothermic vertebrates (fish, lampreys, hagfish, amphibians, reptiles) are integral components of such ecosystems, and therefore can also come into contact with SARS-CoV-2 that has entered the aquatic environment. In this regard, the study of susceptibility to coronavirus of these animals is an urgent task.

To study the tissue and species tropism of SARS-CoV-2, known continuous cell cultures [58], as well as primary cell cultures [59], are used. Despite the low level of expression in fibroblasts of the ACE2 gene (angiotensin-converting enzyme 2) [60], the main receptor for SARS-CoV-2, a number of authors have shown that the virus can effectively infect cells lacking this receptor [61]. In this regard, primary fibroblasts can also be used to study species’ susceptibility to SARS-CoV-2.

For the SARS-CoV-2 virus, the dependence of replication efficiency on temperature in susceptible cell cultures was established [62,63]. Ambient temperature can also affect the efficiency of virus replication in cells of poikilothermic animals. Based on this factor, we evaluated the effect of temperature on the replication of SARS-CoV-2 in the cell lines used at values typical for the period of active life of cell donor animals.

**SARS-CoV-2 infection in reptile cell cultures**. In our study, we used the only continuous cell culture of the eastern box turtle heart (*Terrapene carolina*) TH-1, which was available in Russian collections of cell cultures, obtained in 1967 [64]. Despite the fact that this turtle species is not directly associated with aquatic ecosystems [65], the cell culture obtained from it demonstrated a high susceptibility to various fish and amphibian viruses [64], which suggests the potential sensitivity of TH-1 to viruses present in the aquatic environment. This cell culture is sensitive to various warm-blooded animal viruses, including human viruses [64]. However, the susceptibility of TH-1 to animal and human coronaviruses has not been studied previously. Thus, in order to study the species tropism of SARS-CoV-2, in addition to the continuous cells TH-1, we also took cultures of primary fibroblasts of Nile crocodile (*Crocodylus niloticus*) embryo CNI, which had not been previously used in virological studies.

We have shown that SARS-CoV-2 can effectively replicate in continuous cells and primary fibroblasts of certain reptile species without adaptation. The incubation temperature of infected reptile cells had a significant impact on the course of infection.

It can be noted that the dynamics of change in Ct (accumulation of subgenomic RNA N of the SARS-CoV-2 protein) in reptile cells and Vero E6 at 29 °C, as well as in Vero E6 at 37 °C, were similar (Figure 2C), but the infectious virus titer curves in reptile and Vero E6 cells differed (Figure 2D). The infectious virus titer curves in both susceptible reptile cell lines were characterized by the presence of peaks with maximum values, one of which, related to TH-1, corresponded to the time of onset of CPE (Figure 1), while, in Vero E6, a directly proportional increase in the infectious titer SARS-CoV-2 was observed. This may be due to selective translation of SARS-CoV-2 subgenomic RNAs in non-host cells, as previously described by Rottier et al. when studying the translation of subgenomic RNAs of three strains of A59 mouse coronavirus in oocytes of the clawed frog (*Xenopus laevis*) [66]. TH-1 epithelial-like cells were characterized by a higher susceptibility to SARS-CoV-2 infection compared to primary CNI fibroblasts, which may be due to the significantly greater presence of the main ACE2 receptor on cells of this type.

**SARS-CoV-2 infection in fish cell cultures**. In our experiment, we used a continuous cell culture FHM obtained from the caudal peduncle of the fathead minnow (*Pimephales promelas*) in 1965 [49]. Like TH-1, FHM has also been studied for susceptibility to warm-blooded animal viruses [50], among which is the avian coronavirus, the causative agent of infectious bronchitis (infectious bronchitis virus), currently belonging to the avian coronavirus species, genus *Gammacoronavirus* [53]. Absence of sensitivity for FHM to the infectious bronchitis virus was shown: infection of the cell culture during three successive passages led to a gradual decrease in the infectious titer of the virus up to complete elimination [50]. The susceptibility of FHM to other coronaviruses, including human coronaviruses, has not been investigated. CGIB diploid fibroblasts from the swim bladder of silver carp (*Carassius gibelio*) have not previously been used in virological studies.

The FHM and CGIB fish cell cultures we used showed no susceptibility to SARS-CoV-2 infection, regardless of the virus dose and the incubation temperature. At an infecting dose of the virus, which was the same for fish and reptile cells, no viable SARS-CoV-2 was detected in the culture medium of FHM- and CGIB-infected cells for 2 days after infection, even in trace amounts, in contrast to the primary fibroblasts of the CCAR2f loggerhead turtle. This indicates a less effective interaction of receptors in fathead minnow and silver carp cells with SARS-CoV-2, which is consistent with the results of Xie et al. when studying SARS-CoV-2 infection in cultures of fathead minnow (EPC, skin), grass carp (CIK, kidney), and bluegill sunfish (BF-2, caudal peduncle) cells [42].

Infection of FHM and CGIB with a higher dose of virus revealed differences between them in susceptibility to SARS-CoV-2, which was especially noticeable at different incubation temperatures (Figure 3B,D). At a temperature of 25 °C, SARS-CoV-2 adsorbed on FHM cells remained viable for 3 days and for a day after infection, at 29 °C. CGIB diploid fibroblasts showed low susceptibility to infection with SARS-CoV-2: one hour after infection, a small amount (≤1 log_10_ TCID_50_/mL) of the infectious virus remained both at 25 °C and 29 °C incubation temperatures. Thus, continuous FHM cells can adsorb SARS-CoV-2 on their surface while maintaining its viability under certain conditions. The preservation of the infectious properties of SARS-CoV-2 in various types of water during this time has been shown for temperatures of 20–24 °C [45,67,68], which may be sufficient for the virus adsorption, for example, on the gill epithelium of some fish species. The results obtained show that the ability to adsorb SARS-CoV-2 is significantly higher in epithelial-like FHM cells compared to CGIB diploid fibroblasts, which may also be due to differences in the cell receptor apparatus and a greater presence of ACE2 on FHM.

**Comparison with SARS-CoV-2 infection in Vero E6 cell culture**. It was previously shown that SARS-CoV-2 can replicate without adaptation in susceptible cell cultures, including Vero E6, at temperatures of 33–34 °C with different productivity [62,63]. In the process of assessing temperature adaptation, it is possible to obtain cold-adapted SARS-CoV-2 variants capable of replication at an incubation temperature of 21 °C [52], characterized by a decrease in virulence in vitro and in vivo. The SARS-CoV-2/human/RUS/Nsk-FRCFTM-1/2020 strain used by us showed the ability to replicate in cell culture Vero E6 at 29 °C without adaptation with less productivity than at 37 °C (Figure 2D and Figure 3D). At 25 °C, no infectious virus was detected in the Vero E6 culture medium during the experiment (Figure 2B and Figure 3B). Thus, the Vero E6 cells showed high susceptibility to SARS-CoV-2 at 37 °C, moderate susceptibility at 29 °C, and low susceptibility at 25 °C. In fact, at 25 °C, Vero E6 cells become non-sensitive to infection with wild-type SARS-CoV-2 (non-cold-adapted variants), as do cells from genetically distant hosts (reptiles, fish).

## 5. Conclusions

Currently, the global spread of SARS-CoV-2 is no longer considered a public health emergency of international concern [69]. At the same time, the circulation of the virus among animals and occasional hosts, independent of the human population, ensures the emergence of new variants with unknown and unpredictable infectious properties. We have shown the ability of SARS-CoV-2 to effectively replicate in the cells of certain reptile species without adaptation—in a continuous epithelial-like cell culture of the eastern box turtle heart (*Terrapene carolina*) TH-1 and primary fibroblasts of Nile crocodile (*Crocodylus niloticus*) embryo CNI at 29 °C. In terms of sensitivity to SARS-CoV-2 infection, TH-1 cells incubated at 29 °C were comparable to the reference Vero E6 cell culture incubated at 37 °C. The results obtained by a number of researchers in the successful adaptation of virus to replication at low (21 °C) temperatures in susceptible Vero E6 cells [52] suggest that such adaptation is also possible in reptile cells sensitive to SARS-CoV-2. The findings also provide a basis for special attention to reptiles as occasional susceptible hosts of SARS-CoV-2 in the context of the ongoing global spread of the virus and its continuous evolution.

## Figures and Tables

**Figure 1 viruses-15-02350-f001:**
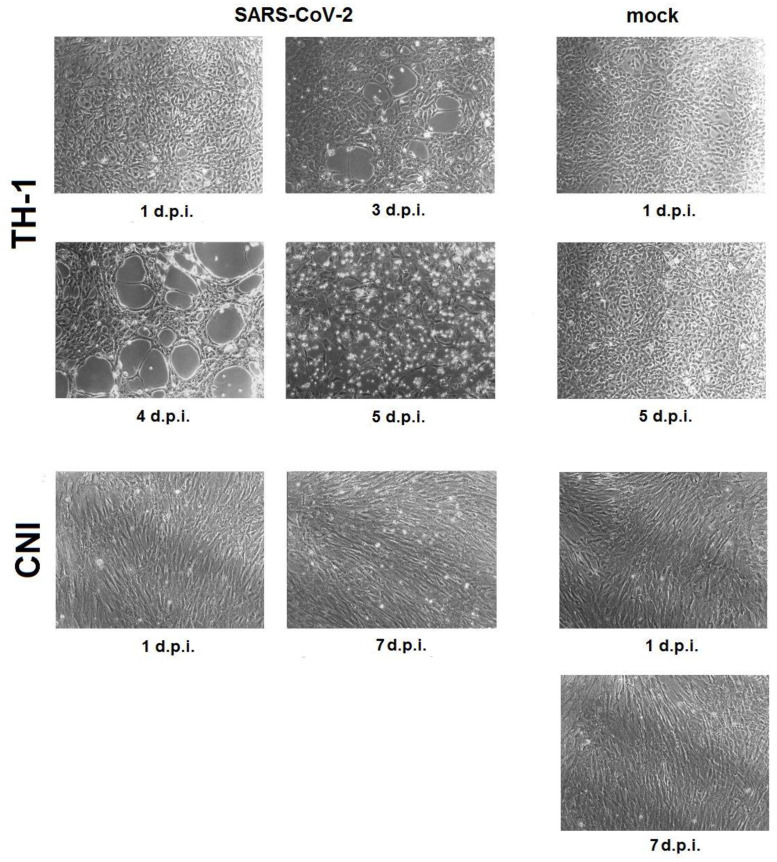
Changes in TH-1 and CNI reptile cell cultures infected with SARS-CoV-2, incubated at 29 °C. d.p.i.—days post infection.

**Figure 2 viruses-15-02350-f002:**
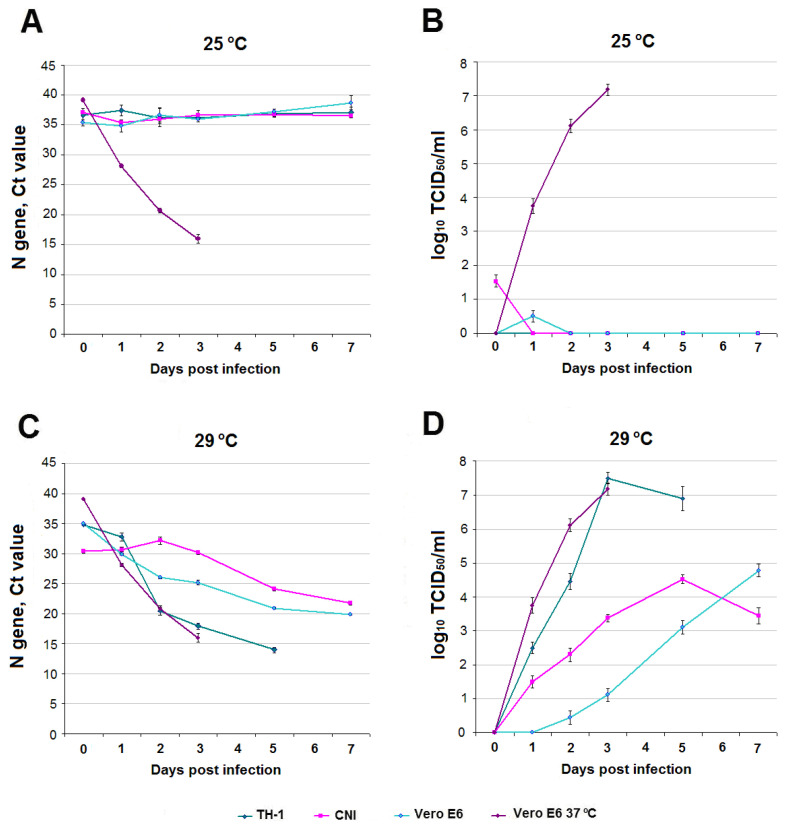
SARS-CoV-2 infection in TH-1, CNI reptile cell cultures, and Vero E6 reference culture. (**A**,**C**)—Ct values at 25 °C and 29 °C. (**B**,**D**)—infectious titer of SARS-CoV-2 log_10_ TCID_50_/mL at 25 °C and 29 °C. For comparison, data on SARS-CoV-2 infection in Vero E6 at 37 °C are presented.

**Figure 3 viruses-15-02350-f003:**
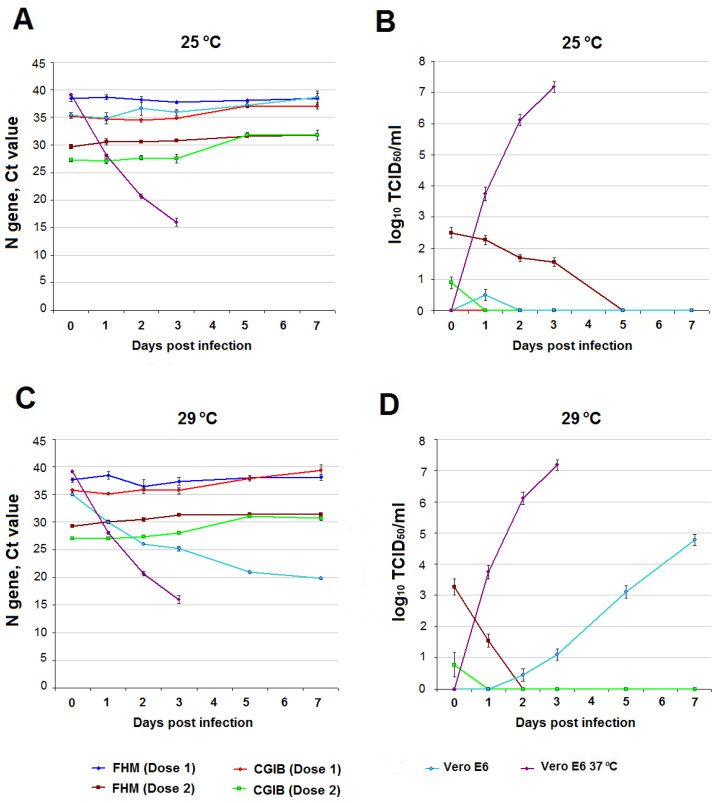
SARS-CoV-2 infection in fish cell cultures FHM, CGIB, and Vero E6 comparison culture over time. (**A**,**C**)—Ct values at 25 °C and 29 °C. (**B**,**D**)—infectious titer of SARS-CoV-2 log_10_ TCID_50_/mL at 25 °C and 29 °C. For comparison, data on SARS-CoV-2 infection in Vero E6 at 37 °C are presented.

**Figure 4 viruses-15-02350-f004:**
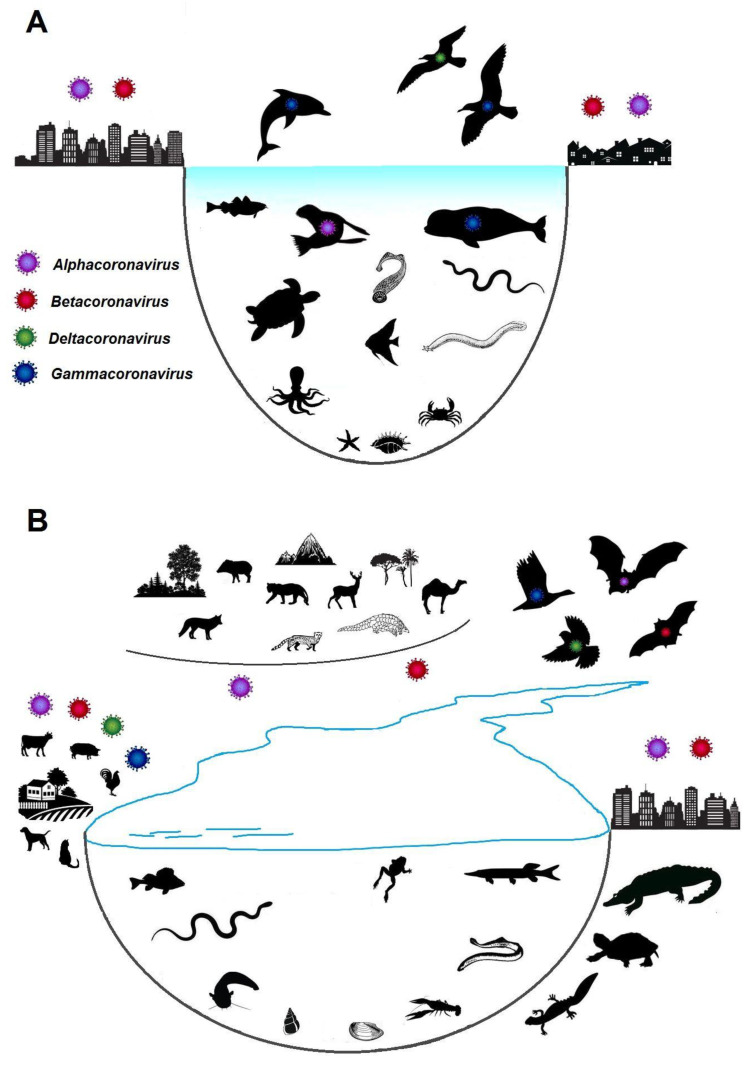
Natural routes of contact of animals in aquatic and coastal ecosystems with coronaviruses. (**A**)—oceanic and marine ecosystems; (**B**)—freshwater ecosystems.

**Table 1 viruses-15-02350-t001:** Cell cultures used in the experiment.

Name of Culture, Organ (Tissue) Origin	Type of Culture	Composition of Growth Medium	Cultivation Conditions	Collection
Vero E6 African green monkey (*Chlorocebus sabaeus*), kidney	continuous, epithelial-like	MEM with L-glutamine, 10% FBS (Fetal Bovine Serum), gentamicin 5 μL/mL	5% CO_2_, 37 °C	State Research Center of Virology and Biotechnology «VECTOR», Rospotrebnadzor
TH-1 eastern box turtle (*Terrapene carolina*), heart	continuous, epithelial-like	DMEM with L-glutamine, 10% FBS, gentamicin 5 μL/mL	24 °C	National Research Center for Epidemiology and Microbiology named after Honorary Academician N F Gamaleya of the Ministry of Health of the Russian Federation
FHM fathead minnow (*Pimephales promelas*), caudal peduncle	continuous, epithelial-like	DMEM with L-glutamine, 10% FBS, gentamicin 5 μL/mL	24 °C	National Research Center for Epidemiology and Microbiology named after Honorary Academician N F Gamaleya of the Ministry of Health of the Russian Federation
CNI Nile crocodile (*Crocodylus niloticus*), embryo	primary fibroblasts, 4th passage	Alpha MEM with L-glutamine, 15 FBS, gentamicin 5 μL/mL	5% CO_2_, 30 °C	Cryobank of Cell Culture, Core Facilities Centre, IMCB, SB RAS
CGIB Silver carp (*Carassius gibelio*), swim bladder	diploid fibroblasts, 22th passage	Alpha MEM with L-glutamine, 15% FBS, gentamicin 5 μL/mL	5% CO_2_, 28 °C	Cryobank of Cell Culture, Core Facilities Centre, IMCB, SB RAS

## Data Availability

The data presented in this study are available in an article here.

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
