# Peer review of "Features of SARS-CoV-2 Replication in Various Types of Reptilian and Fish Cell Cultures"

_viruses, 2023, doi:10.3390/v15122350_

Round 1
Reviewer 1 Report
Comments and Suggestions for Authors
In this manuscript the authors (Kononova et al.) describe their investigations of the ability of a number of fish and reptile derived cell lines to support the replication of SARS-CoV-2, the rational being that species within those animal groups might get exposed to the virus via waste water, sewage, flood run-offs, contact with birds and other species, etc., notably around major cities and from intensive farming enterprises. Employing five such cell lines (three fish-derived, two reptile-derived), they test the cells at different temperatures and with different virus challenge doses (the MOI should be provided rather than or in addition to the inoculum size). Vero cells are used as the comparator, tested at both 34C and the optimal temperature of 37C. As read out, they assess the cells for CPE, as well as conduct RT-PCR for viral RNA in tissue culture supernatant and virus titration on Vero cells. Only a turtle-derived cell line (TH-1), grown at 29C, supports SARS-CoV-2 replication to a level comparable to Vero cells at 37C, with the virus inducing extensive CPE in the cells. Viral replication does also occur to a limited degree in a crocodile-derived cell line, but CPE is not seen. No virus replication is detected in the three fish cell lines.
While these studies do not prove that fish are resistant to SARS-CoV-2, the results may indicate that it would only be in special circumstances that infection would take place and one might question whether it would be sustainable. In contrast, reptilians might – at least under high ambient temperatures – be able to support virus replication, but it remains to be ascertained whether this would happen to a degree, where they could sustain a reservoir in nature such as has been see with white-tailed deer in North-America (PMID: 37640694). Crocodilians, in particular, are known to be able to contribute to a transmission cycle of some human pathogenic viruses, notably arboviruses (e.g., PMID: 35114499, 35632847, 32054016). With climate change and ever-increasing environmental temperatures this could become of some concern.
The Discussion is very long with a lot of reiteration of results, and should be shortened to focus on the main points derived from the study. Also, Figure A1 could be incorporated into the manuscript as Figure 3. Figures 1 and 2 will need improvement to enhance sharpness/focus. Currently they are quite blurry.
The manuscript could be further improved by addressing the following:
· Line 23: delete ‘that’
· Lines 38-40: this sentence needs rephrasing for improved syntax and greater clarity.
· Line 44 & throughout the manuscript: Latin species names should be written in Italics.
· Line 55: correct to “and in nature”
· Line 58: change to “documented in 26 mammalian species ……”
· Line 71: change to "from feces and urine are uncommon” (or infrequent).
· Line 75: change to “and river waters is likely due to low …..”
· Line 91: suggest change to “are likely underestimated”.
· Lines 116-118: this sentence is unclear and should be rephrased.
· Line 41: reference no. 46 is apparently in Russian and the methodology should therefore be described in details here, as it is unlikely that the majority of readers of VIRUSES understand Russian.
· Line 142: change to “All experiments were carried out …….”
· Line 147: change to ‘experiments’
· Table 1: change ‘4 passage’, ‘7 passage’ and ’22 passage’ to either “4th passage” or “passage 4” etc.
· Line 160: since the term ‘fetal bovine serum’ and the abbreviation FBS is used later and in Table 1, then it would be appropriate to also use “fetal bovine serum” here and introduce the abbreviation FBS here rather than in line 165.
· Line 165 and elsewhere in the manuscript: TCID50 should be written with the 50 as subscript.
· Line 256: please reorganize the order of the parentheses. Also, it is unclear why the genus names are repeated in the parenthesis.
· Lines 277-8: change to “with human and animal alpha- and beta-coronaviruses ………..
· Line 288: delete ‘coronavirus’ (the word is already included in the abbreviation).
· Line 304: delete ‘virus’ (the word is already included in the abbreviation).
· Line 315-6: correct to “sensitivity to various viruses of warm-blooded animals”
· Line 339: something seems to be missing between ‘observed’ and ‘SARS’?
· Line 356: delete ‘was’
· Lines 378-84: this is a lot of repetition of details from the Results section and should be cut out or shortened.
· Line 413: correct to “infection, TH-1 cells incubated at 29C were comparable ……”
END
Comments on the Quality of English Language
Please see suggested changes above.
Author Response
Dear Reviewer.
We would like to thank you for all your efforts to make our manuscript better. Attached you will find our response to your comments.

Reviewer 2 Report
Comments and Suggestions for Authors
Summary:
This is an interesting research paper on the ability of SARS-CoV-2 to replicate in reptilian and fish cell cultures. The work provides further evidence in defining the species tropism of the virus. The authors further suggest that these findings could help develop bioassays to detect the presence of infectious virions in river and sewage water samples. The findings of this study are novel and add to the field's understanding. There are issues with the plots and interpretations observed. The quantification and maybe design of assays may need to be repeated.
Suggested action: Major revision
Major comments:
- Lines 71-73, there had been previous successes in isolating infective virions from fecal samples and evidence of transmission related to sewage contamination. Reports were published as early as May 2020 describing the isolation of infectious virions from fecal samples. (10.3201/eid2608.200681) Studies on this topic may be scarce because of the rich microbiological content of fecal samples and sewage, making isolation in cell culture difficult. The authors should summarize earlier work on this topic and the related evidence of transmission attributed to sewage contamination. Examples/ reference: 10.1056/NEJMoa032867, 10.3390/jcm10122696, 10.1016/j.scitotenv.2023.163049.
- Lines 98-101, the sentence is incomplete. The author is correct in distinguishing between the detection of RNA and the presence of infectious particles. They should use more hedging when suggesting the possibility of the virus overcoming the species barrier and going up the food chain.
- Section 2: were the cell lines defined (e.g., cytochrome c sequencing to confirm species) and cultures free of mycoplasma? This is standard practice nowadays and critical in this study.
- Line 167: Why were two virus dosages used for inoculation?
- Figures 2 and 3: Questionable experimental technique. Plots are incomplete and confusing.
- Plotting the Ct value obtained from RT-qPCR assays is not the standard practice. A standard curve should be constructed using a plasmid serial dilution to convert the Ct values to copy numbers, as PCR efficiencies vary between kits, conditions, and batches. They are widely available from the reagent depository or can be cloned easily. The authors are strongly suggested to repeat the quantification.
- Why weren’t all cultures maintained to 7 days post-infection? They seem to terminate at random periods if I’m not mistaken.
- The x-axis is not in linear increments. There are jumps between 3, 5, and 7 dpi, which is not the convention.
- The color of choice for the lines is confusing. The color hue for TH-1 and Vero E6 in 37c is too similar, making reading tricky. The authors should choose a color-blind-friendly palette and use symbols if needed.
- What is lg TCID50/ml expressed throughout the article? Is it the log of TCID50/ml? Is it a log-based 10, 2, or e? The authors are suggested to follow the conventional method, e.g. 1.2 x10^5 TCID50/ml
- In figures 3 B and D, some experimental groups are missing from the plot, explain.
- In figure 3, panels A and C, the 0-dpi data point varies significantly between samples. The most significant difference is more than 10 Ct, which converts to over 1000-fold in copy number. As the authors collected culture medium supernatant right after the adsorption process after washing with buffer twice, the remaining virus in suspension should be minimal and consistent between the various cell lines used. The variation could mess with the actual infection dose and subsequent propagation. The data from RT-qPCR does not correlate with the TCID tittering, which raises further questions.
- Discussion paragraph 2: as the authors previously pointed out, the retrieval of infectious SARS-CoV-2 from rivers and sewage has not been reported earlier, the chances of aquatic animals encountering CoV is relatively low. Flash floods and other events will likely lower the chance, not increase it, as the substantial amount of water flow will dilute the viral load even further. The authors could propose such a theory, but they should be careful when making these suggestions.
- Line 342: on selective translation. To circumvent this, the authors should consider other RT-qPCR assays that target genes further up the 3’ region of the SARS-CoV-2 genome. The possibility of selective translation leading to a discrepancy between TCID and qPCR data is just a theory, the authors should use some hedging. The formation of defective particles or incompatibility in viral packaging and viral release could also be the reason.
- Line 376-389: the titer of the culture supernatant after adsorption and washing should not be used to infer the susceptibility of the cell line. If the authors want to study this, cell monolayer lysate should be collected for RT-qPCR to detect attached and internalized virions. The titer variations post-adsorption and washing in the supernatant may only be an issue on competence as the virus adsorbed should bind firmly or have already been internalized.
Minor comments:
- In lines 44 & 47, italics should be used for the species' scientific name.
- What is lg TCID50/ml expressed throughout the article? Is it the log of TCID50/ml? Is it a log-based 10, 2, or e? The authors are suggested to follow the conventional method of expressing it in its original form, e.g. 1.2 x10^5 TCID50/ml
- Line 256: What’s with the bracket? Is there a mix-up here?
- Discussion paragraph 3: it would be a nice addition to the study if the author would quantify the expression levels of ACE2 in the cells used and their level of similarity with the human ACE2 orthologue. Nucleotide, AA, and structural comparisons can be made. Further protein binding, knock-out, and over-expression assays could be performed to better characterize the interaction.
- Line 337-339: check your writing.
- Line 345-348: is this only a theory or validated by your experiments? Please cite the relevant studies or present the data.
N/A
Author Response
Dear Reviewer.
We would like to thank you for all your efforts to improve our manuscript.
Attached you will find the response to your comments.

Round 2
Reviewer 2 Report
Comments and Suggestions for Authors
the authors have addressed my concerns
Author Response
Dear Reviewer!
We are sorry for misunderstanding of the question.
The X-axis in figure 2 and 3 was corrected. We hope that at this time we dispeled your concerns.
Thank you for you efforts to improve our manuscript.
Sincerely yours,
Authors.